# A deep learning method trained on synthetic data for digital breast tomosynthesis reconstruction

**Arnaud Quillent** [1,2]                                                    ARNAUD.QUILLENT@TELECOM-PARIS.FR
**Vincent Bismuth** [1]
**Isabelle Bloch** [2,3]
**Christophe Kervazo** [2]
**Saïd Ladjal** [2]

[1] *GE HealthCare, Buc, France*
[2] *LTCI, Télécom Paris, Institut Polytechnique de Paris, Palaiseau, France*
[3] *Sorbonne Université, CNRS, LIP6, Paris, France*

**Editors:** Accepted for publication at MIDL 2023

## Abstract

Digital Breast Tomosynthesis (DBT) is an X-ray imaging modality enabling the reconstruction of 3D volumes of breasts. DBT is mainly used for cancer screening, and is intended to replace conventional mammography in the coming years. However, DBT reconstructions are impeded by several types of artefacts induced by the geometry of the device itself, degrading the image quality and limiting its resolution along the thickness of the compressed breast. In this study, we propose a deep-learning-based pipeline to address the DBT reconstruction problem, focusing on the removal of sparse-view and limited-angle artefacts. Specifically, this procedure is composed of two steps: a classic reconstruction algorithm is first applied on normalised projections, then a deep neural network is tasked with erasing the artefacts present in the obtained volumes. A major difficulty to solve our problem is the lack of real conditions artefact-free data. To overcome this complication, we resort to a new dataset comprised of synthetic breast texture phantoms. We then show that our training method and database strategy are promising to tackle the problem as they improve the informational value of planes orthogonal to the detector, which are not currently used by radiologists due to their poor quality. Eventually, we assess the impact of removing the bias components from the network and using stacks of slices as inputs, with regard to the generalisation ability of our approach on both synthetic and clinical data.

**Keywords:** DBT reconstruction, inverse problem, deep learning, limited angle, sparse view, synthetic phantoms, 2.5D.

## 1. Introduction

Digital breast tomosynthesis (DBT) is an X-ray imaging modality used for breast cancer screening (Niklason et al., 1997). It relies on several low-dose cone-beam acquisitions that are performed at different angles. The resulting projections can then, in principle, be used to reconstruct a 3D volume modelling the breast. DBT devices are however impeded by several physical constraints, such as the X-ray source rotation being limited to a certain angular range (limited angle), and the number of projections not exceeding a dozen (sparse view). Consequently, the 3D volume reconstruction quality is generally dramatically reduced.

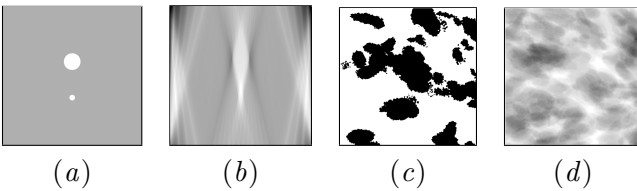

Figure 1: Examples of phantoms. (a) and (b) are coronal slices of a geometric phantom and its FBP reconstruction: as can be seen, the objects are difficult to distinguish in the reconstruction due to limited-angle and sparse-view artefacts along the vertical direction. (c) is an axial slice of a breast texture phantom used in this work and (d) a projection of the whole phantom.

Several reconstruction algorithms have been designed to recover volumes from DBT projections (Sechopoulos, 2013). They can be divided into two main families, the analytical methods (*e.g.*, Filtered Back-Projection – FBP), and the algebraic methods (*e.g.*, Simultaneous Iterative Reconstruction – SIRT). Nevertheless, when applied to projections which are generated by a system suffering from the aforementioned physical limitations, these techniques do not provide high precision reconstructions (Vedantham et al., 2015). The resolution on the source-to-detector axis (vertical direction) is severely limited, and a number of artefacts emerge due to the geometrical constraints (see Figure 1 (a) and (b)): objects are propagated through planes and tend to disappear too slowly, restricting the ability to determine their true boundaries. This spreading effect results in breast tissues being poorly separated along the vertical direction. As a consequence, only the planes parallel to the detector are interpreted by radiologists during an examination.

In this work, we tackle limited-angle sparse-view DBT reconstruction. Specifically, we propose a deep learning reconstruction method for a realistic DBT device with 9 projections acquired over a 25° angular range. To our knowledge, no author has yet proposed a pipeline of this kind to address a geometrical setting as constrained as the one studied here. The method comprehends two steps: simulated data are first processed by a classic analytical reconstruction method, namely FBP, enabling to obtain an initial, albeit imperfect reconstruction of the 3D volume. A deep neural network is then applied in order to correct the artefacts mentioned above. Due to the lack of datasets with ground truth volumes, the network is trained on meticulously crafted reconstructions of synthetic data. Such a 2-step approach is empirically shown to greatly improve the image quality of DBT reconstructions compared to FBP. An overview diagram summarising our method is available in Appendix A. In addition, we further analyse the interest of giving more spatial context as input to the model (2.5D) and investigate the use of bias-free neural networks.

## 2. Related works

Over the last years, several works have been using deep learning to tackle X-ray imaging reconstruction (Wang et al., 2020). Most of these are however dedicated to computed tomography (CT), and only few of them address both sparse-view and limited-angle artefacts.

Two main workflows address X-ray reconstruction, depending on how deep-learning is used inside the reconstruction pipeline (Liang et al., 2019; Wang et al., 2020).

First, some authors rely on a single neural network to perform the whole reconstruction directly from the projections. Nonetheless, merely using generic neural network architectures requires many high-quality training images to obtain good results, and as pointed out before, ground-truth data are not available for DBT. Therefore, some works rely on designing network architectures better taking into account the nature of the problem at hand by *unrolling* iterative reconstruction methods. Deep learning algorithms for DBT reconstruction mainly use this strategy (Wu et al., 2020; Teuwen et al., 2021; Su et al., 2021). Toward the same objective, others use *plug-and-play* approaches, replacing some mathematical operators of an iterative method by a neural network (Liu et al., 2022; Hu et al., 2022). Additionally, hybrid reconstruction pipelines combine two neural networks operating respectively in the projection and volume domains (Liang et al., 2019; Zhou et al., 2022). Although having shown some good results for several inverse problems, both *unrolled* and *plug-and-play* methods yield network architectures that might not be flexible enough to benefit from the whole training set information, as their performance is inherently restricted to specific prior forms (Zhang et al., 2017).

Secondly, some works propose to combine neural networks with classic reconstruction algorithms. In pre-processing pipelines, a neural network (Liang et al., 2019) works in the measurements domain to simulate the missing projections, which are then fed into a classic reconstruction algorithm. In post-processing pipelines (Jin et al., 2017; Qiao and Du, 2022), a first imperfect reconstruction is obtained with a conventional algorithm before being processed by a neural network to remove the artefacts.

## 3. Methodology

### 3.1. Mathematical problem statement

Let $X \in \mathbb{R}^{i \times j \times k}$ be a vector representing a 3D object to image with size $(i, j, k)$, and $Y \in \mathbb{R}^{w \times h \times m}$ the vector of the corresponding $m$ DBT projections on a detector of width $w$ and height $h$. The X-ray acquisition can be modelled as $Y = \mathcal{P}X + \eta$, where $\mathcal{P}$ is the projection operator, and $\eta$ is the realisation of a random variable modelling the noise. DBT reconstruction consists in retrieving $X$ from the noisy measurements $Y$. However, because of the physical constraints set out in the introduction, the measurements $Y$ do not carry enough information to compute a perfect estimation of $X$. Said differently, DBT is an ill-posed inverse problem.

### 3.2. Algorithm

To solve our DBT reconstruction problem, we adapt the model developed by Jin et al. (2017) to address CT reconstruction. This architecture is often used as baseline for posterior works, and we assess how it performs on our DBT dataset. As pointed out in Section 2, the authors of this article propose a post-processing deep learning pipeline on top of a conventional FBP reconstruction step. We use the same approach here and detail in the following of this subsection the deep learning part.

The model is based upon U-Net (Ronneberger et al., 2015), with addition of a residual connection between the input and the output. In the decoder part of the network, we make sure that deconvolution operations consist of an upsampling followed by a convolution to avoid chequerboard artefacts (Odena et al., 2016). The depth of the U-Net is set to 5 levels, with feature maps size increasing from 64 to 1024, as in the original paper. Nevertheless, in contrast to Jin et al. (2017), we study in Section 4 the relevance of removing bias terms from the convolutions and the batch normalisation steps of the model. This adjustment turns the network into a homogeneous function (i.e., $f(\alpha x) = \alpha f(x)$), which has been shown to be more efficient on some inverse problems like denoising (Mohan et al., 2020).

In addition, processing a 3D data volume with the above 2D network requires slicing it as explained in Section 3.3. Such an approach can however lead to deteriorated results, as all the spatial information present in the full 3D volume is not leveraged. Therefore, we propose here to resort to a so called 2.5D approach. 2.5D consists in stacking neighbour slices of the 3D data volume in a sliding window with the aim of inferring the central image. Turning to such inputs can drastically improve the visual aspect of output images, sometimes as much as using heavier 3D convolutions (Ziabari et al., 2018). Besides, implementing this technique requires very few modifications to the initial 2D network. Only the first convolutional layer is altered, where the number of input features is taken as the width of the sliding window. In Section 4, we compare the results obtained with the 2D and 2.5D versions of our model and show that the latter enables improvements.

To train our deep learning model, we use a region of interest (ROI) supervised L2 loss inspired by Wu et al. (2020), yet with a different purpose. In fact, we want to focus the loss on the texture area rather than the padding because the latter does not hold useful information to optimise the network weights. Hence, the ROI is centred on the texture:

$$J = \frac{1}{N} \sum_{i=1}^{N} \|x_i^* - x_i\|_2^2 + \lambda \frac{1}{N} \sum_{i=1}^{N} \|\mathbf{M}_i \odot (x_i^* - x_i)\|_2^2 \, , \tag{1}$$

where $x_i^*$ and $x_i$ are respectively the output slice $i$ and the corresponding ground truth (i.e., the phantom), $\mathbf{M}_i(j,k) = [1 \text{ if } (j,k) \in \text{texture area of } x_i, 0 \text{ otherwise}]$ is a binary mask, $\odot$ denotes the element-wise multiplication, $N$ is the batch size, and $\lambda$ is a hyperparameter to balance the two terms. We observed that removing the whole-image term from the loss function worsens the results, as the network is unable to retrieve the boundaries of the ROI due to the gradients not being updated in the masked-out regions.

### 3.3. Simulated dataset

Training the neural network in a supervised fashion requires data of sufficient quantity and quality. In the following, we create and normalise a novel database to suit our needs.

**Synthetic phantoms** We have at our disposal a stochastic model that can generate synthetic phantoms whose texture mimics the one inside a breast (Li et al., 2022). The 108 phantoms we created are made up of two materials, namely simulating glandular and adipose tissues. The spatial distribution of these textures follows 12 different configurations which are computed from the inner part of real breast CT images. A sample of one of the phantoms is displayed in Figure 1 (c) and (d).

The 108 generated phantoms have a depth and a width of 5 cm, while their heights are taken randomly between 2.5 and 7 cm to mimic a various range of thicknesses. The digital twin of a commercial DBT device is then used to compute the X-ray projections of those phantoms (Sánchez de la Rosa et al., 2019). No noise and no photon scatter are simulated, and phantoms are lesion-free.

**Normalisation step** Pixel values of projections correspond to photon counts. This quantity $I$ can be expressed thanks to the Beer-Lambert law, whose discretised expression is:

$$I_p = I_p^0 e^{-\sum_j \Delta_{pj} \mu_j},$$

where $I_p^0$ is the number of photons emitted by the X-ray source towards the detector pixel $p$, $\mu_j$ is the attenuation coefficient of voxel $j$, the latter being taken along the X-ray beam joining the source to detector pixel $p$, and $\Delta_{pj}$ is the length travelled by the same X-ray beam inside of voxel $j$. Taking the natural logarithm of the projections makes our problem linear. Blank projections (*i.e.*, without any object placed on the detector) are used to estimate $I^0$ as in this set-up, photons only go through air, a material with negligible attenuation.

Images are further normalised by the breast thickness, an information which is known from the acquisition phase, in order to retrieve the mean linear attenuation for each pixel. Besides, photons are colliding at different angles with the detector pixels as this part of the device is static. To compensate for this effect, pixel values are scaled according to their distance to the source.

**Reconstruction** DBT volumes are reconstructed from normalised projections thanks to a FBP-based algorithm. The resolution used to perform this reconstruction is set on all three axes to the detector pixel size (*i.e.*, 100 µm), resulting in isotropic voxels. The volumes are reconstructed with a margin of 10 mm below the breast support and above the phantom height computed during the acquisition phase.

**Data pairing** Our network is trained in a supervised way, and therefore requires a reference image. However, this kind of data must be created thanks to a device with a wide angular range, which is not available in the case of DBT imaging. To overcome this difficulty, we resort to synthetic phantoms whose composition is known and can be used as ground truth. The memory size of full-resolution reconstructed volumes makes them difficult to handle for the creation of a simple proof of concept. Hence, both phantoms and reconstructions are downsampled to a resolution of 500 µm after application of a low-pass filter to avoid aliasing issues. We sample oblique coronal images from the reconstructed volumes along the chest wall to nipple axis, so that the artefacts we want to address are coplanar to the obtained slices. As phantoms are generated with different thicknesses, 2D slices are zero-padded to give them a constant size of $256 \times 256$ pixels. Eventually, we end up with 8 856 coronal images. These slices are then separated into three subsets according to the phantom they are created from. Among the initial 108 phantoms, 24 are selected to be equally divided in a validation and test set. The training set is made up of the remainder of the phantoms. We make sure that each one of the 12 texture sets used for generation is represented in all the sets. Finally, the number of slices in training, validation, and test sets amounts respectively to 6 888, 984 and 984.

Table 1: Evaluation measures computed with 2D models averaged on the test dataset.

| | FBP | 2D | |
| --- | --- | --- | --- |
| | | *Bias-free* | *With bias* |
| **PSNR ↑** | 24.28 ± 1.79 | 30.08 ± 3.41 | **30.10 ± 3.23** |
| **SSIM ↑** | 0.68 ± 0.07 | **0.86 ± 0.06** | **0.86 ± 0.06** |
| **RE ↓** | 0.026 ± 0.007 | 0.009 ± 0.006 | **0.008 ± 0.005** |

Table 2: Evaluation measures computed with 2.5D models averaged on the test dataset.

| | 2.5D, 7 slices | | 2.5D, 13 slices | |
| --- | --- | --- | --- | --- |
| | *Bias-free* | *With bias* | *Bias-free* | *With bias* |
| **PSNR ↑** | 31.51 ± 3.27 | **31.78 ± 3.44** | 31.00 ± 3.41 | 31.08 ± 3.40 |
| **SSIM ↑** | **0.89 ± 0.05** | **0.89 ± 0.05** | 0.88 ± 0.06 | 0.88 ± 0.06 |
| **RE ↓** | **0.006 ± 0.004** | **0.006 ± 0.004** | 0.007 ± 0.004 | 0.007 ± 0.004 |

## 4. Experiments

**Implementation details**   To train our model, we used Adam optimiser with a learning rate of $10^{-4}$, which decreases exponentially after each epoch. Each batch contains 8 images, as we observed that both too small and too large batches have a negative impact on the final results. The training dataset is virtually augmented with vertical and horizontal flips. To further increase the variability of the input images, we also apply a constant random shift to the pixel values. In the end, the training database contains 27 552 images.

**Evaluation measures**   To evaluate our method, we use three measures: peak signal-to-noise ratio (PSNR), structural similarity index (SSIM, Wang et al. (2004)), and relative error (RE). They are computed only in the texture area, with the same mask as in Equation (1). Results are gathered in Tables 1 and 2. Note that due to the lack of methods addressing DBT reconstruction, our approach is compared to FBP only. We can observe that for all configurations of the network, the bias-free version mentioned in Section 3.2 performs slightly worse than the classic one with bias. This behaviour is interesting and demonstrates that even if our model has been empirically shown to be homogeneous, the findings of Mohan et al. (2020) might not be applicable to our problem. Next, we look successively at the results of the 2D and the 2.5D networks, as well as the relevance of bias-free approaches.

**2D reconstructions**   Test set volumes inferred with the best 2D model are displayed in Figure 2. We observe that the breast tissues match pretty well with the ground truth. However, the horizontal edges, where information is most lacking, are the ones where we see the majority of errors. The boundaries between the two materials on the output reconstructions are quite blurred. Indeed, because of its smoothness around 0, the L2 loss term tends to even out the regions having a steep gradient between close pixels. Moreover, the model seems to perform worse on slices with a high heterogeneity of tissues distribution. We think that this is due to the blobs of materials being too small in these images for the network to take them into account. The downsampling step applied during the dataset

creation might be responsible for this phenomenon. Stacking output 2D slices from our model along the chest wall to nipple axis, we can retrieve a 3D volume. Figure 3 shows an example of axial slice taken from this kind of volume. The image displays horizontal hatchings, also observed by other authors (Teuwen et al., 2021). These degradations can be linked to a lack of spatial continuity between 2D slices in the stacking direction. On this basis, we below assess the relevance of 2.5D approaches to lessen these artefacts.

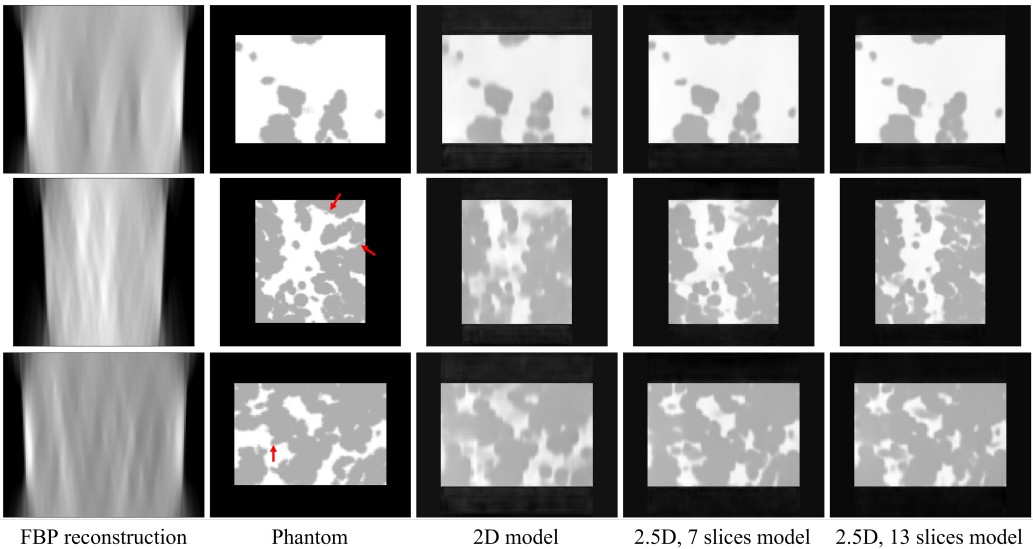

| FBP reconstruction | Phantom | 2D model | 2.5D, 7 slices model | 2.5D, 13 slices model |

Figure 2: Reconstructed coronal slices of 3 different synthetic phantoms from the test set. Grey areas correspond to adipose tissues, white ones to glandular, and black ones to the zero-padding. The arrows are pointing to edges the network struggles to retrieve. Above images are computed with the best model for each configuration.

**2.5D results**  2.5D inputs are a simple way to add continuity without resorting to 3D convolutions (see Section 3.2). Figure 2 shows coronal slices reconstructed with the best 2.5D model. As displayed in Figure 3, hatchings on axial planes are smoothed out, but wrong decisions of the network can be propagated through the volume. Yet, as shown in Tables 1 and 2, all results benefit from more depthwise spatial context. The best performance is obtained with 7 input slices, and additional context does not further improve the results. This demonstrates both the benefits and limitations of the 2.5D approach.

**Test on clinically accurate phantoms**  Unlike synthetic phantoms, clinical DBT images cannot be compared to a ground truth. As we want to assess the performance of our algorithm on this kind of data, we resort to the use of breast CT to create realistic simulations. Indeed, this imaging modality is much less impacted by the sparse-view and limited angle artefacts we want to tackle. First, adipose and glandular tissues of these CT scans are manually segmented. Then, we use these segmentations as phantoms and process them with the same methodology as for the synthetic data. In the end, we have at our disposal

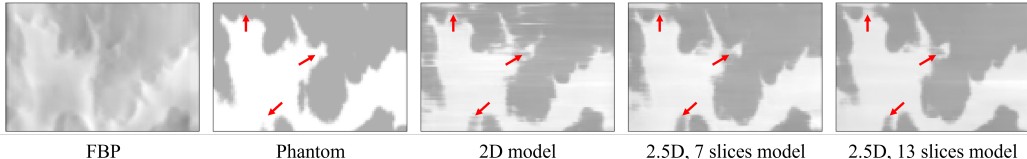

| FBP | Phantom | 2D model | 2.5D, 7 slices model | 2.5D, 13 slices model |

Figure 3: Examples of a reconstructed axial slice. Hatchings are visible on the 3rd image but are smoothed out on the two last images. However, artefacts marked by the arrows are spread.

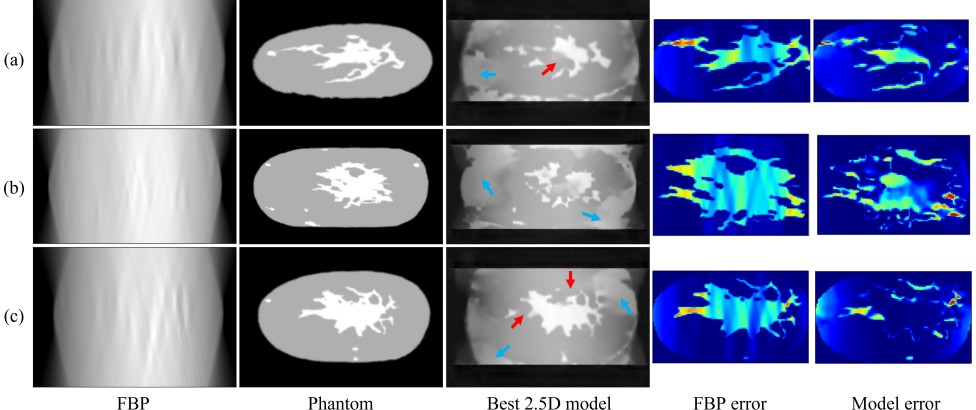

| FBP | Phantom | Best 2.5D model | FBP error | Model error |

Figure 4: Example coronal slices of reconstructions from 3 clinically accurate phantoms. Red arrows point to gaps artefacts, and blue ones to border artefacts. Heatmaps of squared error between the central areas of the phantom and FBP / best model outputs are displayed in the last 2 columns. Axial slices are visible in Appendix B.

FBP reconstructions of objects mimicking real breasts, and a segmentation that can be used as ground truth. Some slices of such images along with their reconstructions with the best 2.5D model are presented in Figure 4. Corresponding evaluation measures are displayed in Table 3. Although the retrieved shapes are relevant, the model creates artefacts at the border of the breast along with gaps inside the tissues. This behaviour is expected as the model tries to reproduce the rectangular shape of phantoms on which it was trained rather than the real ovoid appearance of a breast. To be fair when comparing FBP and the results from our algorithm, we compute the evaluation measures in a central area that contains few border artefacts. Results from our proposed model are more visually appealing than FBP reconstructions, and the associated evaluation measures are higher. Note that these results being exploratory, further analyses are required to ensure that this behaviour is constant over a wide variety of breasts phantoms, differing both in shape and density. Besides, we

Table 3: Evaluation measures of the best 2.5D model computed on the central area of each one of the three images from Figure 4.

|  | (a) | | (b) | | (c) | |
|---|---|---|---|---|---|---|
|  | *FBP* | *Model* | *FBP* | *Model* | *FBP* | *Model* |
| **PSNR** ↑ | 22.24 | 24.48 | 22.11 | 23.94 | 21.40 | 22.10 |
| **SSIM** ↑ | 0.75 | 0.83 | 0.70 | 0.73 | 0.64 | 0.72 |
| **RE** ↓ | 0.039 | 0.023 | 0.036 | 0.024 | 0.048 | 0.041 |

propose in Appendix C an experimental setup to evaluate the quality of real clinical data reconstructions without resorting to a reference image.

## 5. Conclusion

In this paper, we proposed a deep learning pipeline for DBT reconstruction. Great care was used to generate the training set, in particular for the meticulous normalisation step. In addition, we also assessed the utility of 2.5D inputs rather than 2D, and showed that introducing more spatial context results in better reconstructions. Thanks to these improvements, the proposed method reaches good performances, both on synthetic and real datasets.

Our approach might be enhanced with regard to several aspects. First, we strongly believe that our results would be improved using 3D convolutions, as it would give control of the spatial information to the model. Furthermore, as shown by this study, training on realistic synthetic data enables to generalise to clinical images. Developing the realism of the latter, to be even more faithful to a real breast, with a greater variability of materials like fibres and lesions, would doubtlessly be beneficial. Last but no least, the proposed method lacks a parameter to ensure that the projections of the final reconstructed volumes match the ones that were acquired thanks to the device.

## Acknowledgments

This work was partially funded by the French Ministry for Higher Education and Research as part of CIFRE grant No. 2021/1209.

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

## Appendix A. Overview diagram

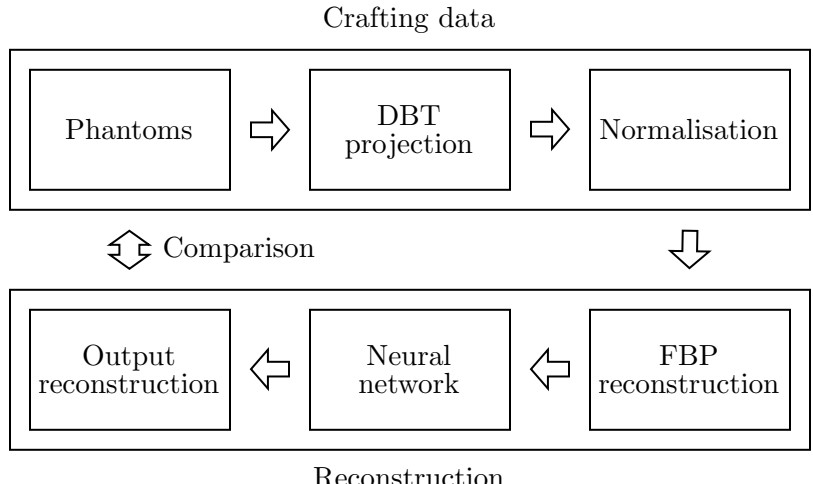

Figure 5: Flowchart representing the proposed reconstruction pipeline.

## Appendix B. Axial slices of reconstructions from clinically realistic phantoms

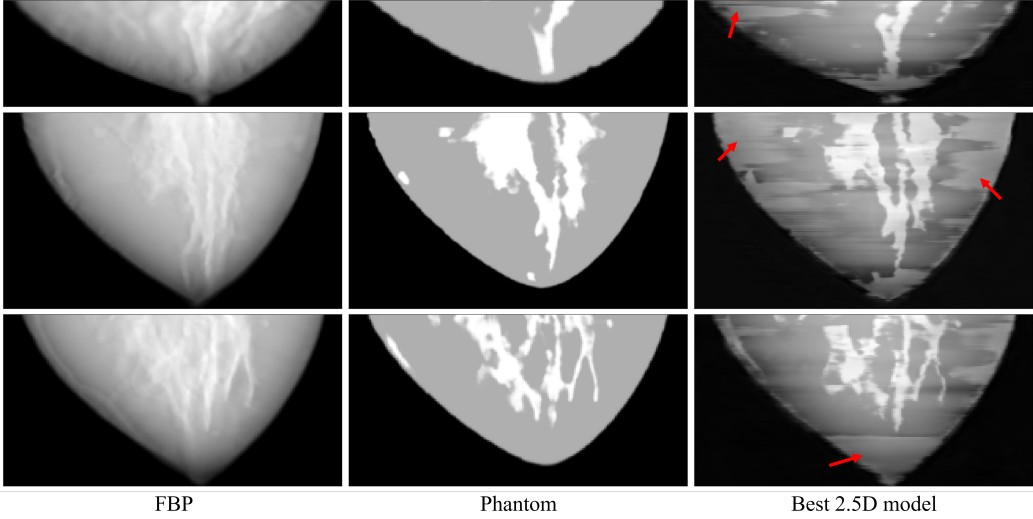

Figure 6: Example axial slices of reconstructions from 3 clinically accurate phantoms. The red arrows correspond to the border artefacts identified in Section 4.

## Appendix C. Experiment on a real clinical image

In this experiment, we applied our pipeline to a real clinical DBT image. Figure 7 tends to show that fibres and gland are better separated with our best model than with FBP. Although trained on synthetic images without lesions, our method is able to reconstruct plausible volumes containing a suspect mass. The hatchings mentioned in Section 4 and Figure 3 are however clearly visible. To assess the capacity of the neural network to reconstruct the depth limits of the lesion present inside this breast, an image quality expert blindly evaluated its dimensions on the FBP reconstruction. We then compare the findings with the visual result outputted by the model (Figure 7). The mass location is precisely recovered by our algorithm, as its reconstructed boundaries lie within the range estimated by the expert.

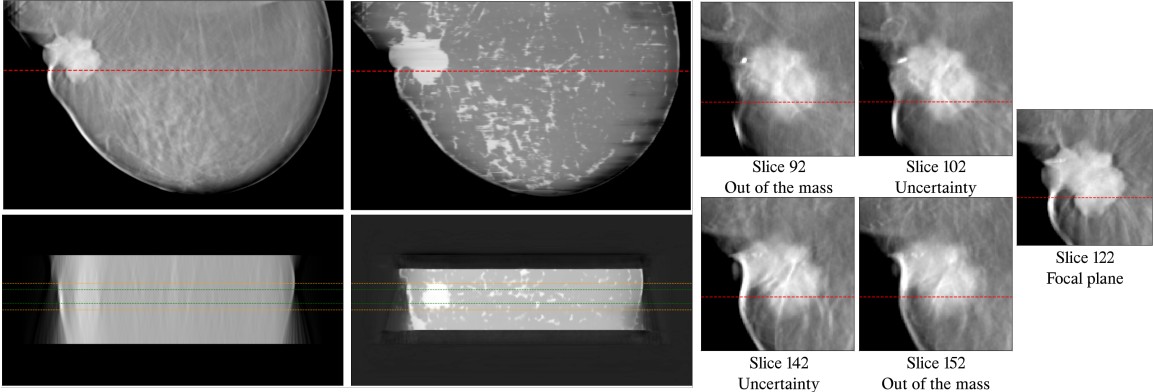

Figure 7: Example reconstruction of a clinical case with a mass using our best model. The two images on the upper left are axial slices of the FBP reconstruction and the result of our best 2.5D model. Bottom left are the corresponding coronal slices taken at the depth of the red dotted line. Orange and green dotted lines in these images are associated respectively with slices at height 102 and 142, and slices at height 112 and 132. Images on the right show FBP axial slices at different heights, with assessment on their actual belonging to the mass.

