# OpenReview forum: "A deep learning method trained on synthetic data for digital breast tomosynthesis reconstruction"
_MIDL.io/2023/Conference — MIDL 2023 Poster_

### Official Review · Reviewer_eqcF · 2023-02-02

**Confidence:** 3
**Preliminary Rating:** 4

**Summary:**

Digital breast tomosynthesis (DBT) is an imaging technique for cancer screening, especially for cancer in dense breast tissue. It enables a 3D (volumetric) reconstruction of the whole breast from a number of low-dose 2D projections obtained by an X-ray tube from different angles. However, the DBT reconstructions are impeded by the geometrical constrains of the DBT device, such as the limited angle rotation of the X-ray source and the sparse view, both leading to decreased image quality and artefacts. To tackle this problem, the authors propose a post-processing supervised DL pipeline on top of the reconstruction step to correct the artefacts present in the obtained reconstructed volumes. The neural network is trained on synthetic images (2D and 2.5D, respectively), mimicking the texture inside the breast. The paper shows that their deep learning pipeline (2.5D network) can reconstruct breast texture phantoms. Testing on a real DBT clinical image, the network is able to reconstruct plausible volumes containing a suspect mass.

**Strengths:**

Focusing on the removal of the geometrical constrains of DBT devices, the combination of a classic analytical reconstruction algorithm with a neural network is a novel approach. Because of the lack of real GT volumes, the authors created a new dataset of synthetic breast texture phantoms. The utility of 2.5D instead of 2D inputs shows that providing more spatial context to the network improves the reconstruction (not a novel strategy per-se, but it seems to be novel for this specific application domain). Figure 1 displays the difficulties of the FBP reconstruction when used only and therefore supports the motivation of the paper.

**Weaknesses:**

- In the conclusion, it is stated, that the authors strongly believe that their approach could be improved using 3D convolutions. Surely, it would make sense to try that since in the clinical application accurate results are more important than computational speed. Therefore, why not compare 2.5D to e.g. 3D instead of 2D to 2.5D if 3D is so promising?
- The post-processing approach was only compared to the FBP-only algorithm but not to other methods. Still unclear: how good is the approach to other methods?
- The additional experiments investigating the removal of bias-free and batch-norm-free neural networks seemed random and not necessary to be included in the framework of the paper.
- Figure 2 is confusing: It is unclear why the FBP reconstructions are not of the same size as the post-processed images of b) to e). The FBP images look stretched. Further, it is unclear which parts of the images display glandular and adipose tissues since they are mentioned in the text but not described in the figure.
- Figure 4 clearly shows the drawbacks of the used training set that is composed of phantoms without a breast shape. Are phantoms with breast shapes complicated to generate?


**Deanonymize Review:**

no

**Detailed Comments:**

- Please add an overview figure of the whole pipeline. Otherwise, it is unclear how the whole procedure was carried out. Including an example of the respective network, inputs would be beneficial.
- Figure 2: How do you compare images with different sizes (FBP vs. post-processed images)? Further, please reduce the black background and display the images bigger. Highlight glandular and adipose tissues – otherwise, it is hard to follow when stated ‘it can be observed that glandular and adipose tissues match pretty well with the ground truth (page 6, 2D reconstructions)
- Figure 4: The five smaller images depict slices that are not easy to follow. More explanation is required to understand why some slices are out of the mass, uncertain, etc. What is the exact purpose or main message?
- Why were *exactly* 108 phantoms generated?


**Paper Type:**

methodological development

**Questions To Address In The Rebuttal:**

As to the loss function defined in equation 1, the authors state that they observed that removing the whole image term would worsen the results and the network would be unable to retrieve the boundaries. But if the first part of the loss function is removed, you only have the L2 norm of two binary functions - which is confusing?
All comments raised under Weaknesses and detailed comments above should be taken into account in a rebuttal, particularly a figure for the pipeline. Here, the additional experiments with the bias-free network could be removed to include the pipeline figure. Based on more explanations and clarifications, the submission would benefit from minor and major suggestions.

---

### Official Review · Reviewer_NyE6 · 2023-02-03

**Confidence:** 3
**Preliminary Rating:** 4

**Summary:**

The authors propose a new method to reconstruct breast tomosynthesis volumes based on a standard reconstruction method followed by a deep neural network to remove artefacts. The method was trained using realistic simulations and virtual phantoms which were lesion free. Evaluation in phantom data shows very good results. The method is also demonstrated in one clinical image.

**Strengths:**

* A well written paper, easy to follow and to reproduce (provided the tools and data are available)
* A sensible approach that seems to provide very good results in virtual phantoms and show promising results in real data

**Weaknesses:**

* the main weakness is the evaluation in just virtual phantom data, and application to a single real image, for which the only analysis is qualitative. More cases from real images with a range of quality and artefact presence would be needed to assess if the method does wrk in real images or not.

* Most methods based on medical image impainting are subject to questioning about how they might deal with anomalies, which is critical for the performance of a medical device. From the result in one real case this does not seem to be an issue but this needs to be proven with more data or discussed in more depth.

**Deanonymize Review:**

yes

**Detailed Comments:**

"In this study, we propose one of the first deep-learning-based pipelines ..." I would remove "one of the first", since it gives little information. Even if we know how many came before, it is not really relevant. The important thing is to propose a method that offers a contribution to the current landscape, and in times where a lot of the research in the field is focusing at throwing DNNs at problems, this is hardly a positive point.



**Paper Type:**

methodological development

**Questions To Address In The Rebuttal:**

I would suggest that authors strengthen the analysis of real cases by using more images, or by discussing further what are the limitations of showing only one image, and describing the anticipated problems that could be caused in real images with varied anatomies, physical properties, and patient demographics.

---

### Official Review · Reviewer_Xnfe · 2023-02-05

**Confidence:** 4
**Preliminary Rating:** 2

**Summary:**

The authors propose a reconstruction pipeline for digital breast tomosynthesis data to remove artifacts. 2.5D strategy and bias-free neural network were used in the proposed method. In the experiments, a simulated dataset is used for training and validation. A real DBT image is used for testing. The proposed method outperforms the classic anatomical reconstruction method.


**Strengths:**

- The study is well motivated that artifact removal is important for the potential deployment of  digital breast tomosynthesis;
- The proposed framework is straightforward;
- The paper is clear and overall well written;


**Weaknesses:**

- In Table, 1, there’s no deep learning based artifact-removal / denoising reconstruction baseline to be compared with the proposed model;
- The proposed method is only validated on one real data;
- There’s no quantitative evaluation of the proposed model on the real data;


**Deanonymize Review:**

no

**Paper Type:**

validation/application paper

**Questions To Address In The Rebuttal:**

- Add denoising reconstruction baseline models for fair comparison;
- Validate the methods on more real clinical digital breast tomosynthesis data;
- Qualitatively evaluate the reconstruction performance of the proposed model on real data;

---

### Official Review · Reviewer_ce68 · 2023-02-06

**Confidence:** 4
**Preliminary Rating:** 4
**Recommendation:** Poster

**Summary:**

The author present a implementation of 2D and 2.5D deep learning model on the reconstruction of digital breast tomosynthesis reconstruction. Unlike some of the related works that replace the classic reconstruction algorithm, this work is a post-processing that works on the reconstructed images by conventional FBP. To address the lack of ground truth for clinial DBT, they created a synthesized dataset with known ground truth. The model trained on the synthetic data seems to transfer very well on the real clinical data.

**Strengths:**

- New application of deep models on the reconstruction/restoration of the DBT.

- The result looks very good on the real data although trained on synthetic images.

- The proposed method is very easy to implement.

**Weaknesses:**

- The idea is very common for many other image restoration problems like denoising. Thus the novelty is very limited.

- The author mentioned several reconstruction methods that combine the neural networks with classic algorithms, but none of them is included as baseline model. Thus, the result evaluation is not strong enough.

- The reconstruction of the real data is hard to evaluate.

**Deanonymize Review:**

no

**Paper Type:**

validation/application paper

**Questions To Address In The Rebuttal:**

For the evaluation of the real clinical images, is it possible to compare with image in other modalities such as mammography? Otherwise the grading from experts should be involved to convince the quality of the synthesized images.

---

### Meta-Review · Area_Chair_UJdr · 2023-02-24

**Recommendation:** Reject
**Confidence:** 3

**Metareview:**

This paper proposes to use a U-net based post-processing step to remove artifacts from digital breast tomosynthesis (DBT) images. The network is trained in a supervised fashion; however, unlike in other applications such as CT where it is possible to collect artifact-free images, e.g. full angle reconstruction, and train to go from images with artifacts, e.g. sparse angle reconstruction, it is not possible to collect artifact-free DBT images. The idea of the paper is to rely on synthetic phantoms (an existing method) to overcome this limitation. As the reviewers pointed out there is no technical novelty in using a U-net as a post-processing step to remove noise/artifacts from medical images. Therefore, the paper is expected to have a strong validation component  I am on the fence whether this paper passes the threshold in this regard. To answer concerns from reviewers regarding this issue, 3 realistic phantom images (generated from CT) were added to the paper; however one can see from comparing Tables 1 and 2 vs 3 that the numerical advantage between the traditional method and the proposed method is now much smaller. This is a very small sample size and I don't think that an argument can be made that in real images that the improvement will be significant. While it might be impossible to test the reconstruction accuracy itself, the authors could consider using a downstream task such as tumor detection to demonstrate improvement with the proposed method. This would be meaningful since the ultimate goal would be clinically relevant downstream tasks performed either by ML models or radiologists reading the reconstructed images.

---

### Meta-Review · Program_Chairs · 2023-03-01

**Recommendation:** Accept (Poster)
**Confidence:** 5

**Metareview:**

The PC discussed this paper, and it was felt that it is an interesting clinical application, especially given the emergence of breast tomosynthesis as the state of the art breast cancer screening modality. Therefore, the decision was to accept as a Poster.